# Casual Sex among Men Who Have Sex with Men (MSM) during the Period of Sheltering in Place to Prevent the Spread of COVID-19

**DOI:** 10.3390/ijerph18063266

**Published:** 2021-03-22

**Authors:** Alvaro Francisco Lopes de Sousa, Layze Braz de Oliveira, Artur Acelino Francisco Luz Nunes Queiroz, Herica Emilia Félix de Carvalho, Guilherme Schneider, Emerson Lucas Silva Camargo, Telma Maria Evangelista de Araújo, Sandra Brignol, Isabel Amélia Costa Mendes, Inês Fronteira, Willi McFarland

**Affiliations:** 1Human Exposome and Infectious Diseases Network, Escola de Enfermagem de Ribeirão Preto, Universidade de São Paulo, Ribeirão Preto 14040-902, Brazil; layzebraz@usp.br (L.B.d.O.); arturqueiroz@usp.br (A.A.F.L.N.Q.); herica_emilly@hotmail.com.br (H.E.F.d.C.); guilherme.schneider@usp.br (G.S.); lucmrg0@gmail.com (E.L.S.C.); iamendes@eerp.usp.br (I.A.C.M.); 2Global Health and Tropical Medicine, Instituto de Higiene e Medicina Tropical, Universidade Nova de Lisboa, 1349-008 Lisboa, Portugal; ifronteira@ihmt.unl.pt; 3Departamento de Enfermagem, Universidade Federal do Piauí, Teresina 64049-550, Brazil; telmaevangelista@gmail.com; 4Departamento de Saúde Coletiva, Universidade Federal Fluminense, Niterói 24220-900, Brazil; sandrabrignol@gmail.com; 5Department of Epidemiology and Biostatistics, University of California at San Francisco, San Francisco, CA 94143, USA; willi_mcfarland@hotmail.com

**Keywords:** casual sex, sex partners, men who have sex with men, SARS-CoV-2, COVID-19

## Abstract

Objectives: We investigated the extent to which Brazilian and Portuguese Men Who Have Sex with Men (MSM) had casual sex partners outside their homes during the period of sheltering in place for the COVID-19 pandemic. Methods: An online survey was conducted in Brazil and Portugal in April, during the period of social isolation for COVID-19, with a sample of 2361 MSMs. Recruitment was done through meeting apps and Facebook. Results: Most of the sample (53.0%) had casual sex partners during sheltering. Factors that increased the odds of engaging in casual sex in Brazil were having group sex (aOR 2.1, 95% CI 1.3–3.4), living in an urban area (aOR 1.6, 95% CI 1.1–2.2), feeling that sheltering had a high impact on daily life (aOR 3.0, 95% CI 1.1–8.3), having casual instead of steady partners (aOR 2.5, 95% CI 1.8–3.5), and not decreasing the number of partners (aOR 6.5, 95% CI 4.2–10.0). In Portugal, the odds of engaging in casual sex increased with using Facebook to find partners (aOR 4.6, 95% CI 3.0–7.2), not decreasing the number of partners (aOR 3.8, 95% CI 2.9–5.9), usually finding partners in physical venues (pre-COVID-19) (aOR 5.4, 95% CI 3.2–8.9), feeling that the isolation had a high impact on daily life (aOR 3.0, 95% CI 1.3–6.7), and HIV-positive serostatus (aOR 11.7, 95% CI 4.7–29.2). Taking PrEP/Truvada to prevent COVID-19 was reported by 12.7% of MSM. Conclusions: The pandemic has not stopped most of our MSM sample from finding sexual partners, with high-risk sexual behaviors continuing.

## 1. Introduction

By the end of December 2020, Brazil was still one of the most affected countries by the COVID-19 pandemic. With more than 200,000 COVID-19 deaths officially confirmed [1], Brazil ranked third in the world in cases of death [2] and the risk of dying from COVID-19 in the country was 3 times greater than in the rest of the world. Portugal, where the infection started spreading nearly one month before Brazil, had relatively early success in controlling the pandemic and had one of the lowest infection rates in the world until the second pandemic wave arrived in spring 2020 [3].

At the end of 2020, however, Portugal experienced an unprecedented worsening of the health crisis by COVID-19. There was an avalanche of new cases, daily records of the number of deaths from COVID-19, and hospitals that were unable to meet all demands. The country ended January with 5576 deaths, a number that represents almost half (44.6%) of the total deaths caused by the new coronavirus since March 2020, when the pandemic started in the country [3].

Despite the arrival of the vaccine in the two countries in December 2020, general preventive measures for respiratory infections (use of masks, hand sanitizer, social distancing, and quarantine) remain the main form of containing the spread of the virus in Brazil and Portugal. Minimizing the gathering and movement of people, to varying degrees of strictness have been adopted by many countries, including Brazil and Portugal [4,5], although in few regions of these countries there were legal provisions that forced people to stay at home (lockdown). There is evidence of the positive effects of sheltering on reducing the spread of COVID-19 infection. However, other aspects of health, including mental and sexual health, may be put aside [6,7].

In face of the COVID-19 pandemic, the In_PrEP group conducted an online survey, seeking to measure the potential consequences of this context on the sexual behavior of men who have sex with men (MSM), in Brazil and Portugal. This group is formed by researchers from Brazilian and Portuguese institutions and analyses the mental and sexual health of MSM from both countries. This research aimed at observing whether MSM were seeking casual partners during the period of the shelter in place directives and which measures they were adopting to reduce the risk of the COVID-19 disease, HIV, and other sexually transmitted infections (STIs). Brazil and Portugal were selected as they share the language and have a large flow of people between these countries each year (28,210) [8], because of immigration, professional and academic activities, and tourism [9].

## 2. Materials and Methods

### 2.1. Study Design, Population, Sampling, and Recruitment

This project was entitled “40tena” (from the Portuguese “quarentena”, which means quarantine, i.e., the period of isolation imposed on those who have had contact with the virus) and is derived from the In_PrEP cohort study, a multicenter survey started in 2020, which carries out behavioral follow-ups of MSM in Brazil and Portugal. This cohort study focuses on investigating issues related to STIs and MSM and is carried out in all 26 Brazilian states and the Federal District and 15 of the 18 districts of Portugal. Due to the COVID-19 pandemic, questions regarding this subject were added to the original project to analyze the sexual behavior of the studied population during the sheltering in place period. The research project and the presentation of these manuscripts were guided by the STROBE tool and The Checklist for Reporting Results of Internet E-Surveys (CHERRIES).

A rapid and dynamic data collection process took place in April 2020 at a time when the two countries were under sheltering directives. In both countries, the directives of the official health agencies asked the inhabitants to “shelter in place”, avoiding close contact with people outside their place of shelter as much as possible. Essential activities such as trade and some services were maintained, but with restrictions.

In total, 2361 MSM participated in the research, out of which 1651 (69.9%) were from Brazil and 710 (30.1%) from Portugal. The participants were recruited by an adaptation of the “snowball” sampling adapted to the virtual environment, so that one participant was responsible for recruiting other individuals of the same category as his, using their networks. To meet the requirements of the method, we selected 15 MSM with different characteristics: location in the country (divided according to the regions of the two countries); race/skin color (Caucasian and non-Caucasian), age (young, adult, and older adult), and schooling level. These were the first participants and they were identified as seeds. Each participant received the link to the research website and was instructed to invite/disseminate to MSM of their social network, until obtaining a significant sample.

The seeds were identified by the two most popular geolocation-based dating applications (Grindr and Hornet) worldwide and in both countries [10] by direct chat with online users, using an adaptation of the time location sampling technique (TLS) to access all regions of the two countries following previous methods [11,12] (Figure 1).

We also used the social network Facebook^®^ to boost the number of participants, directing to the MSM population aged between 18 and 60 years (age limit imposed by the social network), by a fixed post on the official search page (https://www.facebook.com/taafimdeque/ (accessed on 5 March 2021) and it was followed by an electronic link, which provided access to the free and informed consent form and the survey questionnaire.

We included only individuals who identified themselves as men (cis or trans), aged 18 or over, and living in one of the two countries. Non-Portuguese speakers and tourists were excluded. For Facebook recruitment, the researchers used the boost feature to target MSM in both countries.

### 2.2. Measures

Data were collected by Computer-Assisted Technique Interview (CASI). The data collection questionnaire was hosted on the study site and, for security reasons, it only allowed one answer by IP (internet protocol), that is, only one user answered by machine (computer, cell phone.). The questionnaire was created and validated (face-content) by a group of three experts from both countries in two versions: European Portuguese (Portugal) and Brazilian Portuguese (Brazil). Content validity was determined using the content validity coefficient (CVC) using an ordinal scale. The experts were asked questions regarding clarity, objectivity, and relevance, whose answers were graded from 1 to 4 using a Likert scale. It was not suggested to exclude any questions, only changes in the writing according to the country.

The questionnaire was divided into four sections with 46 questions, mostly multiple-choice, some of which were mandatory for the completion of the questionnaire. The questions addressed:Sociodemographic information (age; gender identity; schooling; race/skin color; sexual orientation; type of current relationship; country; state; place of dwelling);Social welfare and coping in the period of social distancing;Sexual practices and activities during the pandemic, namely: sexual practice with a casual partner; sex with the use of drugs; threesome or group sex; sex without condom use; STIs protection strategies; sex frequency and protection measures against COVID-19; search for health services; testing and test result for COVID-19 (self-reported);Sexual practices and activities in the period prior to the pandemic: Use of PrEP and PEP; commonly used ways to search for partners; knowledge about STIs and HIV testing.

For this study, we defined sex with a casual partner or simply casual sex, such as anal sex with a new or unknown partner who was not previously in the same shelter as the participants, with the question: “Since the social distancing/sheltering was proposed in your country, have you had sex with a new or unknown partner who is outside the place where you are sheltered or have you left your shelter to meet that partner?”

### 2.3. Analysis

Descriptive analysis was performed for key numerical and categorical variables. Bivariate and multivariable logistic regression was used to characterize associations with having casual sex. We tested the multicollinearity between variables before moving on to the regression model. A final model was selected, using the forward conditional input method, based on retaining those variables with *p* < 0.1 while using *p* < 0.05 as the cut-off for significance. The best performance of the multivariate model was considered for aspects of accuracy, sensitivity to specificity (Receiver Operating Characteristic—ROC) demonstrating that the statistical performance developed was better than chance. The fit of the models was assessed by the Hosmer and Lemeshow test.

### 2.4. Ethical Considerations

The research project obtained ethical approval from the Universidade Nova de Lisboa, Portugal and Universidade de São Paulo, Brazil. Informed consent was obtained from all users online, before proceeding with the questionnaire.

## 3. Results

A total of 2361 MSM participated in the online surveys, including 1651 (69.9%) from Brazil and 710 (30.1%) from Portugal (Table 1). Most participants had access to the survey from the indication by referral through social networks/partners or colleagues (72.7%), and Facebook^®^ was responsible for the remaining 27.3%. The median age was 29 years (ranging from 18–66). Most men in both countries lived in urban areas (69.0% in Brazil, 95.4% in Portugal) and were single (69.2% in Brazil, 82.3% in Portugal). One in ten (9.9%) MSM respondents in Brazil self-reported their HIV status as positive, as did 12.1% of respondents in Portugal. In Brazil, 10.5% reported testing, and 5.5% reported being diagnosed with COVID-19. In Portugal, 15.5% had tested and 1.8% were diagnosed with COVID-19.

Table 1 also describes the sexual behavior of MSM during the sheltering in place period to prevent the spread of COVID-19. Over half of respondents (1252/53.0%) had casual sex, considering that 47 (3.8%) engaged in paid sex, group sex (27.3%), sex under the influence of alcohol or drugs (69.8%), and unprotected sex (47.1%).

Many MSM reported behaviors that they believed would reduce the risk of COVID-19 transmission. Apart from measures taken concerning sex, general preventive measures (e.g., use of face mask until the meeting place and hand sanitizing with hand sanitizer) (25.8%), asking if the partner was sheltering (30.7%), and asking if the partner had symptoms (27.5%) were mentioned. Other measures to reduce the spread of COVID-19 included avoiding kissing during sex (16.2%), washing hands before and after sex (27.6%), and cleaning the place where they had sex before and after it (14.6%). Notably, among the 652 users of PrEP/Truvada in this study, almost half (301; 46.1%) also stated using this medicine as a preventive measure to COVID-19 transmission.

Table 2 presents correlates of leaving the house or having someone in their house for casual sex during the sheltering period in bivariate and multivariable logistic regression models for each country. In Brazil, the odds of engaging in casual sex increased with the variables having group sex (adjusted odds ratio (aOR) 2.1, 95% CI 1.3–3.4), living in an urban area (aOR 1.6, 95% CI 1.1–2.2), feeling that sheltering had average (aOR 2.2, 95% CI 1.5–3.2) or high impact on their daily life (aOR 3.0, 95% CI 1.1–8.3) compared to low impact, having casual partners (aOR 2.5, 95% CI 1.8–3.5), and not decreasing the number of partners during the COVID-19 epidemic (aOR 6.5, 95% CI 4.2–10.0). In Portugal, the odds of engaging in casual sex increased with the variables using Facebook to find partners (aOR 4.6, 95% CI 3.0–7.2), not decreasing the number of partners during the COVID-19 epidemic (aOR 3.8, 95% CI 2.9–5.9), usually finding partners in physical venues (pre-COVID-19) (aOR 5.4, 95% CI 3.2–8.9), feeling that the isolation had a high impact on their daily life (aOR 3.0, 95% CI 1.3–6.7), and reporting HIV-positive serostatus (aOR 11.7, 95% CI 4.7–29.2).

## 4. Discussion

Our results showed that the COVID-19 pandemic and the period of sheltering in place did not stop a considerable portion of Brazilian and Portuguese (53%) MSM from seeking and finding casual sexual partners. The implications of this are direct contact with partners outside their place of isolation and who have unknown exposure history of coronavirus infection. This behavior can contribute to the chain of infection of SARS-CoV-2. In Portugal, compliance in the general population appears to have been high enough to prevent overwhelming the hospital system [13] at the beginning of the pandemic. However, the continuity of sexual behaviors that imply physical encounters between participants may explain, in part, the second severe pandemic wave that caused the country’s health system to be challenged in late December 2020. On the other hand, Brazil is facing one of the worst COVID-19 health crises to date and sees its mortality rates steadily increase while the portion of the population that has maintained social distancing and other primary infection control measures decreases [14].

Although slightly over half of MSM still found casual partners outside their homes, 75% had fewer partners than before the COVID-19 epidemic. This evidence highlights that the problem is not an increase or decrease in casual partners, but rather the fact that when encounters occur, they seem to be permeated with unsafe sexual practices (unknown partners, drug use, orgies, and use of ineffective prevention methods) increasing vulnerability to SARS-CoV-2 exposition and STIs simultaneously [15].

MSM reported other measures to reduce the risk of COVID-19 akin to other risk-reduction practices. For example, over 30% asked if the partners were sheltering, and more than 25% asked about any symptoms of COVID-19. Although close contact was inherent or implied in having casual sex, many MSM reported avoiding kissing, handwashing before and after sex, and cleaning the place where they had sex before and after it. This report reveals that, although there is exposition, MSM try to manage the possibility of infection by adopting some measures, even if of little efficiency.

An unexpected finding was the use of PrEP/Truvada for COVID-19 prophylaxis. In the absence of evidence of efficacy for COVID-19 prevention, the assumption may lead people on PrEP to neglect effective measures. A possible explanation for the adoption of this practice might be misunderstanding the discussion of the potential of prophylaxis drugs for SARS-CoV2 in the popular media [16]. Some MSM may have mistaken Truvada, promoted for HIV prophylaxis, as having a similar mechanism for SARS-CoV-2. Specific messaging may be needed to dispel this false connection through programs promoting PrEP for HIV.

Our study also found continuing behaviors that may place the studied group at high risk of acquiring or transmitting HIV and other STIs during the COVID-19 epidemic, at levels similar to those reported in studies [10,17] slightly prior to the pandemic period. More than one in six MSM reported group sex, implying the meeting of several people in very close contact, thus amplifying potential COVID-9 exposure [18]. Engaging in group sex was further associated with increased odds of having casual partners among the group of Brazilian MSM. Sexual encounters under the influence of drugs or alcohol, also common during the sheltering period, can decrease reasoning capacity and hinder the adoption of preventive measures for HIV/STIs and COVID-19 [19]. Unprotected sex with a new casual partner was reported by over one in three Brazilian MSM and one in five Portuguese MSM during the shelter in place period. These have been considered high rates of incidence [12].

The duration of the sheltering period, with accompanying feelings of isolation, may partly explain the high-risk sexual behaviors. Most participants had been isolated for at least 30 days, and many recognized a high impact of social isolation on their lives. This in turn may have led MSM to feel a greater need for social contact, to seek a “break” in isolation to seek partners [20], with an additional break for HIV preventive measures. This hypothesis is corroborated by the findings of the multivariable analysis, in which acknowledging the high impact of the sheltering period was associated with seeking external casual sex partners both in Brazil and Portugal. The effect of a prolonged isolation period is particularly worrisome as Brazil moves towards becoming a COVID-19 epicenter in Latin America and the world [21].

Some studies imply that social isolation may lead to higher utilization of virtual networks to search for sexual encounters [6,22]. Connections via the Tinder application increased 15% in the US and 25% in Italy and Spain during the COVID-19 epidemic [22]. The duration of chat activity also increased by 30% [22]. Notably, the use of Facebook as the preferable way to find partners before the pandemic was significantly associated with greater odds for Portuguese MSM having casual sex. Another hypothesis is that partnering through Facebook can provide a false sense of exposure control, as it enables sex with a friend, acquaintance, or otherwise someone belonging to the same social network Yet another possible explanation for the association of increased casual partnering during COVID-19 and the use of Facebook, not yet documented in the literature, may be fear of judgment (i.e., for breaking sheltering) by closer friends, which leads MSM to seek out like-minded strangers. On the other hand, social media can assist with keeping smaller social groups as well as in the adoption of practices such as virtual sex and masturbation [23,24], which may reduce risks of transmissible infections.

Other significant associations with seeking casual sex during COVID-19 are notable. Being HIV positive also increased the odds of engaging in casual sex in Portuguese MSM. Although we recognize that even before the pandemic, studies indicated that MSM living with HIV already had high-risk sexual behavior [25,26], another hypothesis may be in the false sense of protection due to antiretrovirals for HIV currently being tested in COVID-19 patients [27]. This may be consistent with assumptions or misunderstandings about PrEP, as mentioned above.

In both Brazil and Portugal, living in an urban area increased the odds of casual sex, likely explained by access to greater numbers of MSM [28,29]. This facilitates the location and selection of partners by dating applications or other social media [12].

This study has several limitations. First, we recognize the data derived from a convenience sample in both countries, therefore, our results cannot be generalized to all MSM in the countries, although we have created a mechanism for allocating seeds according to the main social and demographic characteristics that cause confusion. Venue-based and peer-referral mechanisms to sample and recruit are harder to logistically accomplish during the COVID-19 epidemic. The site that hosts the form is not able to inform how many subjects were reached, only the amount of access and answers.

Secondly, we did not measure variables recognized as important in hindsight, such as an exact number of days in sheltering, different sexual practices, and the organization of other events, such as parties where sex may have occurred. Lastly, we did not test for COVID-19 and therefore could not fully link behaviors directly to the infection. Furthermore, most questions refer to past events that may be biased by memory.

## 5. Conclusions

We were able to identify a high frequency of casual sex among MSM, as well as associated factors that might increase exposure to SARS-CoV-2, HIV, and other STIs during a period of high COVID-19 transmission after implementation of sheltering in place. Although many strategies were adopted to minimize the exposure to SARS-CoV-2, the effectiveness of those measures is threatened by high-risk practices for both COVID-19 and HIV, including unprotected sexual intercourse and group sex. By analyzing two countries with different scenarios for control of COVID-19, we were able to demonstrate the vulnerability of MSM communities to the pandemic. Our results should be considered when making decisions about public health, since if these vulnerabilities are left unaddressed, they may hamper the response to the pandemic.

## Figures and Tables

**Figure 1 ijerph-18-03266-f001:**
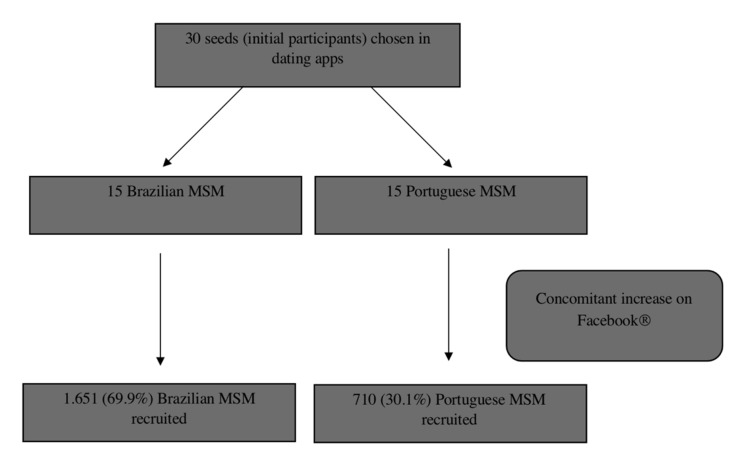
Flowchart for data collection.

**Table 1 ijerph-18-03266-t001:** Characteristics and sexual practices during the COVID-19 shelter in place period, men who have sex with men, Brazil and Portugal, 2020.

	Brazil (N = 1651)		Portugal (N = 710)		Total (N = 2361)	
Characteristics	**n**	**%**	**n**	**%**	**n**	**%**
Gender identity						
Cisgender man	1637	99.2	697	98.2	2334	98.9
Transgender man or non-binary	14	0.8	13	1.8	27	1.1
Schooling						
<9 years	340	20.6	152	21.4	492	20.8
>9 years	1311	79.4	558	78.6	1869	79.2
Race/skin color						
Caucasian	375	22.7	575	80.9	950	40.2
Non-Caucasian	1276	77.3	135	19.1	1411	59.8
Lives in urban area	1140	69.0	677	95.4	1817	77.0
Relationship status						
Single	1143	69.2	584	82.3	1727	73.1
Monogamous	480	29.1	86	12.1	566	24.0
Polyamorous	28	1.7	40	5.6	68	2.9
Self-reported HIV status						
HIV negative	1285	77.8	488	68.7	1773	75.1
HIV positive	163	9.9	86	12.1	249	10.5
I do not know	203	12.3	136	19.2	339	14.4
Tested for COVID-19	174	10.5	110	15.5	284	12
Diagnosed with COVID-19	90	5.5	13	1.8	103	4.4
Are you now sheltering in place?						
No	74	4.5	26	3.7	100	4.2
Partially	405	24.5	154	21.7	559	23.7
Yes	1172	71.0	530	74.6	1702	72.1
For how long have you been sheltering?						
1 to 14 days	60	3.6	54	7.6	114	4.8
15 to 29 days	331	20.1	62	8.8	393	16.7
30 to 45 days	1035	62.7	326	45.9	1361	57.6
More than 45 days	225	13.6	268	37.7	493	20.9
How would you rate the impact that sheltering has had on your life?						
Low impact	215	13.0	70	9.9	285	12.1
Average impact	643	38.9	292	41.1	935	39.6
High impact	793	48.0	348	49.0	1141	48.3
Usual type of sex partner						
Casual	1155	70.0	413	58.2	1568	66.4
Steady	291	17.6	40	5.6	331	14.0
Both casual and steady	205	12.4	257	36.2	462	19.6
Lives with sex partner	236	14.3	109	15.4	345	14.6
Usual ways respondent finds sex partners before period of sheltering						
Dating apps	1285	77.8	544	76.6	1829	77.5
Facebook, Twitter, or Instagram	560	33.9	286	40.3	846	35.8
Other sites	446	27.0	173	24.4	619	26.2
Bars, clubs, bathhouses, cruising areas	72	4.4	25	3.5	97	4.1
Does not search for partners	292	17.7	98	13.8	390	16.5
Decreased number of sexual partners during sheltering	1253	75.9	515	72.5	1768	74.8
In this sheltering period, would you say that…						
Your sexual frequency						
Decreased	1188	72.0	615	86.6	1803	76.4
Did not change	364	22.0	66	9.3	430	18.2
Increased	99	6.0	29	4.1	128	5.4
Your interaction with social media						
Decreased	117	7.1	248	34.9	365	15.5
Did not change	357	21.6	136	19.2	493	20.9
Increased	1177	71.3	326	45.9	1503	63.6
Your alcohol consumption						
Decreased	705	42.7	384	54.1	1089	46.1
Did not change	608	36.8	212	29.9	820	34.7
Increased	338	20.5	114	16.0	452	19.2
During sheltering, the respondent						
Had casual sex	875	53.0	377	53.1	1252	53.0
Sought to pay for sex	63	3.8	9	1.3	72	3.0
Had sex with 2 or more people at the same time	259	15.7	113	15.9	372	15.8
Had sex under the influence of drugs or alcohol	777	47.1	143	20.1	920	39.0
Had unprotected anal sex with a new/casual partner	576	34.9	142	20.0	718	30.4
To protect from COVID-19, the respondent						
Took general/basic protective measures (e.g., using a face mask and hand sanitizer)	423	25.6	187	26.3	610	25.8
Asked if the partner was sheltering	513	31.1	212	29.9	725	30.7
Asked if the partner had symptoms	452	27.4	197	27.7	649	27.5
Avoided kissing during sex	219	13.3	164	23.1	383	16.2
Washed hands with soap and water for at least 20 s before and after sex	450	27.3	202	28.5	652	27.6
Cleaned the place where they had sex before and after sex	209	12.7	136	19.2	345	14.6
Used PrEP/Truvada	191	11.6	110	15.5	301	12.7
Used a condom for anal sex	403	24.4	114	16.1	517	21.9
Did not adopt any strategy	610	36.9	247	34.8	857	36.3

**Table 2 ijerph-18-03266-t002:** Factors associated with having casual sex during the COVID-19 shelter in place period, men who have sex with men, Brazil and Portugal, 2020.

Factors	Categories	OR	95% CI	aOR	95% CI
**Brazil**					
Sought to pay for sex					
	No	1			
	Yes	2.7	1.5–4.8	0.4	0.2–1.1
Sex with ≥2 at the same time (group sex)					
	No	1			
	Yes	10.0	6.6–15.1	2.1	1.3–3.4
Live in urban area					
	No	1			
	Yes	1.4	1.1–1.7	1.6	1.1–2.2
Self-reported impact of sheltering on daily life					
	Low	1			
	Average	1.2	0.9 -1.5	2.2	1.5–3.2
	High	1.1	0.8–1.4	3.0	1.1–8.3
Type of sex partner(s) usually:					
	Steady partner	1			
	Casual and steady partner	3.3	2.2–4.8	1.6	0.9–2.8
	Casual partner	1.5	1.2–2.0	2.5	1.8–3.5
Did you use a condom during sex?					
	No	1			
	Yes	0.4	0.3–0.5	0.6	0.4–0.9
Decrease in the number of partners during sheltering					
	Yes	1			
	No	21.3	15.0–30.4	6.5	4.2–10.0
**Portugal**					
Usually used Facebook to find partners					
	No	1			
	Yes	3.0	2.2–4.2	4.6	3.0–7.2
Did not seek partners					
	No	1			
	Yes	0.5	0.4–0.7	0.3	0.1–0.5
For how long have you been in isolation?					
	15–29 days	1			
	30–45 days	0.5	0.3–0.8	0.2	0.1–0.4
	>45 days	0.8	0.4–1.4	0.4	0.2–0.8
	Not in isolation	0.5	0.3–1.1	0.2	0.1–0.8
Decrease in the number of partners during sheltering					
	Yes	1			
	No	1.2	0.9–1.6	3.8	2.9 – 5.9
Usually found partners at bars, clubs, bathhouses, etc					
	No	1			
	Yes	2.3	1.6–3.3	5.4	3.2 – 8.9
Impact of isolation on daily life:					
	Low	1			
	Average	0.8	0.5–1.4	0.7	0.3–1.7
	High	3.1	1.9–5.4	3.0	1.3–6.7

Self-reported HIV status					
	Negative	1			
	Positive	10.4	4.9–22.0	11.7	4.7–29.2
	Does not know	0.9	0.8–1.1	1.4	0.7–2.3

## Data Availability

All data can be made available, as long as a request is made to the responsible researcher (AFLS).

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
