# Peer review of "Casual Sex among Men Who Have Sex with Men (MSM) during the Period of Sheltering in Place to Prevent the Spread of COVID-19"

_ijerph, 2021, doi:10.3390/ijerph18063266_

Round 1

Reviewer 1 Report

See attached file. 

Author Response

Question 1: State earlier in the introduction the reason why you chose the two counties: Brazil and Portugal were selected as they share a language and a large flow of people between 60 these countries each year (28,210)[8], through immigration, professional and academic activities, and tourism[9].

Answer 1: Thank you very much for that tip. We put the reasons why the two countries were chosen, more related to the development of the In_PreP project and the immigration between the countries

Question 2 - Briefly explain what the In_PrEP Group is the first time you mention it in the introduction. You might want to mention HIV risk in the introduction since I assume In_PREP is a group or research project that was already set up for HIV prevention interventions across the two countries. It’s good to have a paragraph in the introduction for each major issue and population you are addressing. It’s good the introduction is brief, but you then have room to add a couple more paragraphs.

Answer 2: We added this information the first time the project was cited. We added in detail what questions were added to assess the phenomenon COVID-19

Question 3: Grammar or typo in this sentence made it hard to understand:

Answer : We apologize for this error. We carry out the correction and revision of English.

Question 4: What is “40tena”—what does it stand for or what is the translation?

Answer 4 - We added this information: "This project was entitled “40tena” ( from the Portuguese “quarentena”, which means quarantine, i.e.,the period of isolation imposed to those who have had contact with the virus)"

Question 5 - The discussion seems to hit on a lot of the points I raised for the introduction. Therefore, you might want to just add more to the introduction without necessarily making the introduction long.  You might more explicitly state your interpretation of why there were differences in results between the two countries or not. 

Answer 5- Thank you very much. We added more information to the introduction without making it too long, and also more explicitly put our interpretation of why there were differences in results between the two countries or not.

Question 6 - 90 MSM were diagnosed with COVID-19? How does that compare to the rest of the country?

-Answer: Unfortunately, we tried to fit this topic in the discussion, but we did not find it relevant, because besides being self-reported, it is extremely low and different between the two countries.

We are very grateful for all the super important suggestions from reviewer 1. We are available for new requests and suggestions.

Reviewer 2 Report

The work is very interesting

Table 1 should be revised

Author Response

Question 1: Table 1 should be revised

Answer: We reviewed table 1 as suggested.

We appreciate the suggestion of reviewer 2

Reviewer 3 Report

This is a timely, well written article that covers an important area of the research literature on sexual practices of MSM. I mention below a number of ways in which the various sections of the article should be clarified and strengthened.

INTRODUCTION: Overall, the background for the study is well presented. Some clarification with respect to the identity/definition of the In_PrEP Group and their interest in collecting the data would be useful for readers new to this literature.

METHODS: There is strength in the sample size (2,361 MSM) and the recruitment strategy is appropriate for a convenience sample such as this. Why “40tena” as the project title? I am curious about what it means and how it is pronounced.  I suspect others will be as well. There is a good range of interview questions given the Research Question. However, the measure (question) used for Casual Sex appears to be two questions in one. In future research, break it down into two different questions OR pick the most suitable version given what is being measured. also, what about unknown partners coming into the respondent’s place of shelter?

Two small issues: At line 91 there is a reference to Figure 01. There is no Figure 01 in the paper so this must be part of the reference.  If so, make this clearer.  At line 134 the word ‘proving’ is used. Given that social scientists can’t ‘prove’ things to be so, the words 'indicating' or 'demonstrating' would be more appropriate.

RESULTS: For the most part, the discussion of Table 1 is fine.  The Table itself is a problem. First, the columns in the PDF version that I downloaded are too narrow. As a result, the data for each variable appears in two rows. It took me quite a while to figure this out. Second, the table is spread over 5 pages and the headings do not carry over. Third, only a small number of the variables presented in the table are discussed in the text. I recommend that the table be broken into 3 sections with each section reflecting the discussion that is in the text (e.g., paragraph one (lines 141-150), paragraph two (lines 151-154) and paragraph 3 (lines 158-166). If you don’t discuss the descriptive data from the table in the text then it should be in an appendix.  Finally, at line 151 you claim that Table 1 describes how the COVID-19 epidemic ‘changed’ the respondents’ sexual behaviour.  Since you do not have before and after behaviour, the data do not demonstrate this to be the case? Statement needs revising and clarification.

Table 2 and its discussion should be revised in a similar manner.

DISCUSSION/CONCLUSION: The discussion and conclusion are supported by the results and well written. In addition, the limitations of the study are clearly identified with hints of how the study could be improved.

Author Response

Question 1 - INTRODUCTION: Overall, the background for the study is well presented. Some clarification with respect to the identity/definition of the In_PrEP Group and their interest in collecting the data would be useful for readers new to this literature.

Answer 1: Thank you very much for that observation. This suggestion was also given by the reviewer 1, for this reason we have added this information.

-Question 2:  Why “40tena” as the project title? I am curious about what it means and how it is pronounced.  I suspect others will be as well.

Answer 2: Thank you very much for that observation. We added this information: "This project was entitled “40tena” ( from the Portuguese “quarentena”, which means quarantine, i.e.,the period of isolation imposed to those who have had contact with the virus).

Question 3: There is a good range of interview questions given the Research Question. However, the measure (question) used for Casual Sex appears to be two questions in one. In future research, break it down into two different questions OR pick the most suitable version given what is being measured. also, what about unknown partners coming into the respondent’s place of shelter?

Answer 3-Thank you very much for this great and very important suggestion. In fact, our writing was not conveying the full meaning we would like. We seek to question whether the participants "broke through" the isolation / distance / quarantine period. That is, if they had sex contact with someone who had not been with them in their shelter since the beginning. For this reason it may seem that we had two questions, but the final objective was one. We believe it is clearer in the current format.

QUESTION 4- Two small issues: At line 91 there is a reference to Figure 01. There is no Figure 01 in the paper so this must be part of the reference.  If so, make this clearer.  At line 134 the word ‘proving’ is used. Given that social scientists can’t ‘prove’ things to be so, the words 'indicating' or 'demonstrating' would be more appropriate.

Answer 4: Thank you very much again for that observation. We attached an image, but apparently it was not attached. That way, we put it back in the body of the text. In addition, we correct the word indicated, changing "prove" to "indicating"

Question 5. For the most part, the discussion of Table 1 is fine.  The Table itself is a problem. First, the columns in the PDF version that I downloaded are too narrow. As a result, the data for each variable appears in two rows. It took me quite a while to figure this out. Second, the table is spread over 5 pages and the headings do not carry over.

-Answer 5: Due to the rules of the journal, the table really gets huge. We tried to correct the formatting and decreased more than 1 page of the table.

Question 6: Third, only a small number of the variables presented in the table are discussed in the text. I recommend that the table be broken into 3 sections with each section reflecting the discussion that is in the text (e.g., paragraph one (lines 141-150), paragraph two (lines 151-154) and paragraph 3 (lines 158-166). If you don’t discuss the descriptive data from the table in the text then it should be in an appendix. 

Answer 6-: Thank you very much for that suggestion. Due to the authors' writing style, we chose not to consider this change. It turns out that the focus of our text is the factors associated with casual sex, with the other characteristics for presenting the study sample, as well as the factors that were taken to the logistic regression analysis. They are not much data, and to discuss in detail each one of them would imply a giant manuscript, which would hardly catch the attention of the reader. For this reason, we decided to stay focused on discussing the associated factors. We do not put the table as an attachment, as it is essential to understand what factors were tested in the multifactorial analysis / logistic regression

Question 7 -Finally, at line 151 you claim that Table 1 describes how the COVID-19 epidemic ‘changed’ the respondents’ sexual behaviour.  Since you do not have before and after behaviour, the data do not demonstrate this to be the case? Statement needs revising and clarification.

Answer 7 - We fully agree with the reviewer. For this reason, we modified this section. Thanks: "Table 1 also describes the sexual behavior of MSM during the sheltering in place period to prevent the spread of COVID-19". 

We are grateful for the excellent comments and suggestions of the reviewer 3. A number of points raised were extremely important to improve the manuscript.

Reviewer 4 Report

This manuscript entitled "CASUAL SEX AMONG MEN WHO HAVE SEX WITH MEN (MSM) DURING THE PERIOD OF SHELTERING IN PLACE TO PREVENT THE SPREAD OF COVID-19" aimed to verify the extent to which Brazilian and Portuguese MSM had casual sex partners outside their homes during the period of sheltering in place for the COVID- 19 pandemic.

The manuscript is very interesting. However, some issues should be addressed by the authors:

ABSTRACT

  • Line 15: describe the full name for MSM in the first time.

INTRODUCTION

  • Line: 35: Please updated the number of deaths.
  • Last paragraph: Study aim is not clear. Moreover, rationally should be improved. 

METHODS

  • Line 105: Please improve the description of the questionnaire validation process. Please, include what have changed in the questionnaire during this process.

RESULTS

  • In the table 1 you bring information about "testes for COVID" and "Diagnosed with COVID", however this information and how was given this answer is not in the Method section. Please clarify this.
  • From the same concern, do the authors have collected any information about what type of test the participants made? Because we know that different tests have differents capacity to correctly identify the virus.

REFERENCES

- In my opinion several recent articles from COVID 19 from the IJERPH could be cited. Please, go through the recent issues.

Author Response

Question 1: Line 15: describe the full name for MSM in the first time.

-Thanks for that suggestion. We made that change.

Question 2: Line: 35: Please updated the number of deaths.

-Thanks. We updated according to the study reference

Question 3: Last paragraph: Study aim is not clear. Moreover, rationally should be improved. 

-We seek to improve the writing of our goal. Thanks

Question 4: Line 105: Please improve the description of the questionnaire validation process. Please, include what have changed in the questionnaire during this process.

-We added more information as suggested by the reviewer

Question 5: In the table 1 you bring information about "testes for COVID" and "Diagnosed with COVID", however this information and how was given this answer is not in the Method section. Please clarify this.

-We have added this information in the methods section.

Question 6: From the same concern, do the authors have collected any information about what type of test the participants made? Because we know that different tests have differents capacity to correctly identify the virus.

-We did not collect information about type tests, as we understood that not all participants would be able to answer this question. For this reason, this information is not deepened in the text or placed as a causal variable.

Question 7: In my opinion several recent articles from COVID 19 from the IJERPH could be cited. Please, go through the recent issues.

-Since most databases (WoS and SCOPUS) punish journals due to self-citations, we chose not to follow this guidance, to avoid hearing the IJERPH.

We are grateful for the relevant comments from reviewer 4, which were important to improve the manuscript at crucial points,

Round 2

Reviewer 4 Report

All my comments were correctly addressed by the authors.